# Broccoli Extract (Broccoli NMN^®^) Improves Skin Hydration by Regulating HAS and NF-κB Pathways and Reduces Wrinkle Formation via the TGF-βR1/Smad3/Collagen Pathway

**DOI:** 10.3390/cimb48010050

**Published:** 2025-12-30

**Authors:** Wonhee Cho, Yeonhwa Lee, Minhee Lee, Jeongjin Park, Yuki Mukai, Dae Soo Lim, Hyelin Jeon, Woojin Jun

**Affiliations:** 1Research Institute of Medical Nutrition, Kyung Hee University, Seoul 02447, Republic of Korea; wonhi1117@khu.ac.kr; 2Division of Food and Nutrition, Chonnam National University, Gwangju 61186, Republic of Korea; qazwsx917@naver.com (Y.L.); pjj8425@hanmail.net (J.P.); 3Department of Food Innovation and Health, Kyung Hee University, Yongin 17104, Republic of Korea; miniclsrn@khu.ac.kr; 4Research Institute for Human Ecology, Chonnam National University, Gwangju 61186, Republic of Korea; 5AL-FOODS Co., Ltd., 3-2-12 Mita, Tokyo 108-0073, Japan; mukai@al-foods.com; 6Daedeok Pharma Co., Ltd., Suwon 16229, Republic of Korea; ddpmhwk@hanmail.net

**Keywords:** Broccoli extract, Ultraviolet B, oxidative stress, skin photoaging, skin health

## Abstract

This study investigated the effects and mechanisms of broccoli extract containing more than 99.0% β-NMN (BRC) on UVB-induced skin damage, including moisture loss, oxidative stress, inflammation, wrinkle formation, and melanin production, using in vitro and in vivo models. BRC treatment significantly alleviated UVB-induced skin dehydration, oxidative stress, and inflammatory responses, as well as inhibited wrinkle formation and melanin synthesis. Mechanistically, BRC enhanced skin hydration and barrier function by upregulating hyaluronic acid synthases and genes related to sphingolipid metabolism, while simultaneously suppressing NF-κB signaling and COX-2 expression, thereby re-ducing inflammation. Moreover, BRC promoted collagen synthesis by activating the TGF-βR1/Smad3/Collagen pathway and prevented extracellular matrix degradation by inhibiting JNK/c-Fos/c-Jun/MMPs signaling. In addition, BRC modulated the cAMP/PKA/CREB/MITF/TRPs pathway, leading to reduced melanin production. These findings suggest that BRC supplementation may effectively protect against UVB-induced skin damage, supporting its potential application as a functional ingredient for skin health.

## 1. Introduction

The skin, as a key element of the innate immune system and the body’s largest protective organ, not only serves as a physical barrier against external environmental factors but also performs vital physiological functions including thermoregulation, moisture retention, and immune regulation. Structurally, the skin consists of three distinct layers—the epidermis, dermis, and subcutaneous fat layer—each with specialized structural and physiological roles. The epidermis contains keratinocytes that prevent moisture and heat loss, melanocytes responsible for melanin synthesis, Langerhans cells and T lymphocytes involved in immune defense, and Merkel cells associated with sensory perception. The dermis is composed primarily of extracellular matrix (ECM) components, including collagen, elastin, proteoglycans, and glycoproteins synthesized by fibroblasts, which provide mechanical strength and elasticity [1,2]. Ultraviolet (UV) radiation is one of the major environmental factors that significantly impact skin health. UVA (320–400 nm) and UVB (290–320 nm) penetrate the dermis and epidermis, respectively, causing various forms of skin damage [3]. One of the most evident acute effects of UV exposure is inflammation. In particular, UVB irradiation induces excessive production of reactive oxygen species (ROS), including superoxide anions, hydrogen peroxide, singlet oxygen, and hydroxyl radicals, which disrupt the antioxidant defense system and lead to oxidative stress [4,5,6,7]. Moreover, UV exposure triggers a cascade of cytokines, vasoactive, and neuroactive mediators, collectively promoting an inflammatory response and ultimately resulting in sunburn. Oxidative stress induced by UVB exposure contributes to excessive melanin production in epidermal melanocytes and a reduction in hyaluronic acid synthesis in keratinocytes, both of which play a critical role in skin pigmentation and moisture retention [8,9]. Furthermore, in the dermis, UVB-induced oxidative stress stimulates the degradation of extracellular matrix (ECM) proteins by matrix metalloproteinases (MMPs), leading to wrinkle formation, while also promoting the production of inflammatory cytokines [7]. Broccoli extract containing 99% β-NMN (BRC) was investigated for its potential benefits on skin hydration, barrier function, pigmentation, and wrinkle formation. Nicotinamide mononucleotide (NMN), a derivative of vitamin B3 and a precursor of nicotinamide adenine dinucleotide (NAD+), plays a crucial role as an essential coenzyme in cellular metabolism, DNA repair, and mitochondrial function. Upon oral intake, NMN is rapidly absorbed and converted into NAD+, contributing to cellular energy production and metabolic homeostasis [10]. Previous studies have shown that NMN supplementation can reduce age-related inflammation, enhance insulin activity, improve mitochondrial performance, and support neural and physical functions, suggesting its value as an anti-aging nutraceutical. Emerging evidence also indicates that NMN may influence skin pigmentation [11,12].

Despite these findings, the molecular mechanisms by which NMN influences skin hydration and barrier integrity remain unclear. Given the natural, food-derived origin of BRC and its high NMN purity, this study aimed to investigate the underlying mechanisms of skin hydration, skin elasticity, and melanin synthesis using both in vitro and in vivo models.

## 2. Materials and Methods

### 2.1. Preparation of Broccoli Extract (BRC)

Broccoli extract (BRC) was obtained from Daedeok Pharma Co., Ltd. (Suwon, Republic of Korea) and was originally produced as a food ingredient by AL-Foods Co., Ltd. (Tokyo, Japan). Fresh broccoli (*Brassica oleracea* var. italica Plenck) was washed and dried, followed by yeast-mediated fermentation with *Saccharomyces cerevisiae* at 30 °C for 16 h in purified water to enhance NMN production. The fermented material was concentrated under reduced pressure and extracted with food-grade ethanol (95%, *v*/*v*), then subjected to sequential purification steps including filtration, ultrafiltration, nanofiltration, and ion-exchange chromatography to remove insoluble materials and impurities. The purified fraction was vacuum-dried and sterilized, yielding a final product containing ≥99.0% β-nicotinamide mononucleotide (β-NMN), as verified by internal quality control analyses. The BRC was supplied for research use and used in all experiments.

### 2.2. Cell Culture and Treatment

HaCaT, HS27, and B16F10 cells used in this study were obtained from the American Type Culture Collection (ATCC, Manassas, VA, USA). The cells were cultured in Dulbecco’s Modified Eagle Medium (DMEM) supplemented with 10% fetal bovine serum (FBS) (Hyclone Laboratories, Logan, UT, USA), 100 mg/L penicillin-streptomycin (Hyclone Laboratories), 2 mmol/L L-glutamine (Hyclone Laboratories), and sodium pyruvate (Hyclone Laboratories). All cells were maintained at 37 °C in a humidified atmosphere with 5% CO_2_.

For UV irradiation experiments, HaCaT and HS27 cells were exposed to UVB at a dose of 50 mJ/cm^2^ using a UV lamp equipped with five Sankyo Denki G5T5 tubes (Sankyo Denki Co., Yokohama, Japan). The UVB dose (50 mJ/cm^2^) was determined based on preliminary optimization experiments (25–75 mJ/cm^2^), which identified 50 mJ/cm^2^ as the optimal level that induced sublethal oxidative stress without causing excessive cell death in HaCaT and HS27 cells. Prior to the main experiments, the cytotoxicity of BRC was evaluated in HaCaT, HS27, and B16F10 cells using the MTT assay. No significant cytotoxicity was observed at concentrations ranging from 50 to 200 µg/mL. Accordingly, 200 µg/mL was selected as the maximum concentration for subsequent in vitro experiments. Following UVB exposure, cells were treated for 24 h with either L-ascorbic acid (positive control, Sigma-Aldrich, St. Louis, MO, USA; 100 µg/mL) or BRC at concentrations of 50, 100, and 200 µg/mL. For melanin production assays, B16F10 cells were pretreated with 250 µM isobutylmethylxanthine (IBMX, Sigma-Aldrich) to induce melanogenesis, followed by treatment with either arbutin (100 µg/mL, Sigma-Aldrich) or BRC at concentrations of 50, 100, and 200 µg/mL for 72 h.

### 2.3. Animals and UVB Irradiation

SKH-1 hairless mice (five-week-old, male) were obtained from SaeRon Bio (Uiwang, Republic of Korea). The mice were housed in an animal facility with automatically controlled conditions, maintaining a temperature of 23 ± 2 °C, humidity of 50 ± 5%, and a 12 h light/dark cycle. After one week of acclimatization, the mice were randomly divided into six groups (eight mice per group) based on their body weight measurements taken during the adaptation period. The animals were provided ad libitum access to a standard diet (AIN-93G diet, Research Diets Inc., New Brunswick, NJ, USA) and drinking water. This study was conducted following the approval of the Animal Ethics Committee of Kyung Hee University (KHGASP-24-055).

The experimental groups were classified as follows: Normal control (−UVB), Control (+UVB), Positive control (+UVB with dietary supplementation of L-ascorbic acid at 100 mg/kg bw), BRC10 (+UVB with dietary supplementation of BRC at 10 mg/kg bw), BRC30 (+UVB with dietary supplementation of BRC at 30 mg/kg bw), and BRC60 (+UVB with dietary supplementation of BRC at 60 mg/kg bw). To induce photoaging, the dorsal skin of the mice was exposed to UV radiation from a UV lamp (Sankyo Denki Co.) three times per week for eight weeks. The minimal erythemal dose (MED) was set at 150 mJ/cm^2^, with exposure levels progressively increasing as follows: Week 1: 1 MED (150 mJ/cm^2^), Week 2: 2 MED (300 mJ/cm^2^), Week 3: 3 MED (450 mJ/cm^2^), and from Week 4 to Week 8: 4 MED (600 mJ/cm^2^). At the end of the eight-week experiment, the mice in each group were sacrificed, and blood and skin tissues were collected for further analysis.

### 2.4. Measurement of Transepidermal Water Loss (TEWL)

The hydration level of the dorsal epidermis was measured using a moisture analyzer (Howskin, Seoul, Republic of Korea) under controlled conditions of 23 ± 2 °C and 50 ± 5% humidity. The device’s electronic probe was placed in contact with the UVB-irradiated dorsal skin of the mice for measurement. This assessment was conducted in the eighth week of the experiment.

### 2.5. Morphological and Histological Observations

To observe morphological changes, close-up images of the dorsal skin of the mice were captured using a digital camera. For histopathological analysis, the dorsal skin of the experimental animals was excised, flattened onto filter paper, and fixed in 10% neutral formalin before being embedded in paraffin. The paraffin-embedded tissues were then sectioned into 4 μm-thick slices using a microtome (RM21225, Leica, Wetzlar, Germany) and subsequently stained with hematoxylin and eosin for histological examination.

### 2.6. Measurement of Antioxidant Enzyme Activity

The supernatant of HaCaT cells and dorsal skin tissues from experimental animals were collected to assess the activities of superoxide dismutase (SOD), catalase (CAT), and glutathione peroxidase (GPx). These activities were measured using the SOD Assay Kit, Catalase Assay Kit, and Glutathione Peroxidase Activity Assay Kit (Abcam, Cambridge, UK). The absorbance was recorded at 655 nm using an ELISA reader (Bio-Rad Laboratories, Hercules, CA, USA).

### 2.7. Protein Extraction and Western Blot Analysis

Dorsal skin tissues and cells were lysed in lysis buffer (Sigma-Aldrich, St. Louis, MO, USA) containing a protease inhibitor to extract proteins. A total of 30 μg of protein samples was separated by sodium dodecyl sulfate-polyacrylamide gel electrophoresis (SDS-PAGE, 10%) and transferred onto a nitrocellulose membrane (Bio-Rad Laboratories). The membranes were blocked for 1 h in TBS-T buffer (0.5% Tween 20 in TBS buffer) containing 5% skim milk powder. Subsequently, they were incubated at room temperature for 3 h with primary antibodies against CerS4 (LASS4), p-/IκBα, p-/NF-κB, COX-2, p-/JNK, p-/c-Fos, p-/c-Jun, p-/Smad3, Collagen I, pro-Collagen I, MMPs (-1, -2, -3, -9), p-/PKA, p-/CREB, MITF, TRPs (-1, -2), and β-actin (Cell Signaling Technology, Beverly, MA, USA). After incubation with HRP-conjugated secondary antibodies (Cell Signaling Technology) for 1 h at room temperature, protein bands were detected using an enhanced chemiluminescence (ECL) system (Amersham Pharmacia Biotech, Piscataway, NJ, USA) and imaged using Easy Photo. The Western blot band images were quantified using ImageJ software ver.1.54k (NIH, Bethesda, MD, USA).

### 2.8. Total RNA Isolation and Real-Time Polymerase Chain Reaction (PCR)

HaCaT and HS27 cells were collected using 0.25% trypsin-EDTA, and total RNA was extracted using the RNeasy Mini Kit (QIAGEN, Hilden, Germany). The extracted RNA was then reverse-transcribed into cDNA using the iScript™ cDNA Synthesis Kit (Bio-Rad Laboratories). For gene amplification, real-time quantitative PCR was performed using SYBR Green (iQ SYBR Green Supermix, Bio-Rad Laboratories) with the CFX Connect™ real-time PCR detection system (Bio-Rad Laboratories). The nucleotide sequences of the primers used for gene amplification are listed in Table 1. For skin tissue samples, the excised tissues were collected and dissolved in 1 mL TRIzol (QIAGEN), followed by the addition of 200 μL chloroform to separate the aqueous phase. RNA was then extracted using the RNeasy Mini Kit (QIAGEN) and converted into cDNA using the iScript™ cDNA Synthesis Kit (Bio-Rad Laboratories). The sequences of the primers used for gene amplification are provided in Table 2.

### 2.9. Measurement of Hyaluronic Acid and Sphingomyelin

HaCaT cells were lysed using 1% Triton X-100 in phosphate-buffered saline (PBS), and the levels of hyaluronic acid and sphingomyelin were measured using the Hyaluronic Acid ELISA Kit (Abcam) and Sphingomyelin Assay Kit (Abcam), respectively, following the manufacturer’s protocols. Absorbance was recorded using an ELISA reader (Bio-Rad Laboratories).

### 2.10. Measurement of Pro-Inflammatory Cytokines

The secretion levels of interleukin (IL)-1β, IL-6, and tumor necrosis factor-alpha (TNF-α) were measured using the Duoset ELISA Kit (R&D Systems, Minneapolis, MN, USA) according to the manufacturer’s protocol. The analysis was performed using an ELISA reader (Bio-Rad Laboratories) with the supernatant from HaCaT cell cultures.

### 2.11. Measurement of Intracellular Melanin and Tyrosinase Activity

To measure melanin content, B16F10 cells were lysed and treated with a solution containing 1N NaOH and 10% DMSO, followed by incubation at 100 °C for 10 min. The absorbance was then measured using an ELISA reader (Bio-Rad Laboratories). For tyrosinase activity measurement, B16F10 cells were lysed and analyzed using the Tyrosinase Activity Assay Kit (Abcam) according to the manufacturer’s protocol. The absorbance was recorded using an ELISA reader (Bio-Rad Laboratories).

### 2.12. Measurement of Nitric Oxide, Glutathione, and cAMP Levels

The supernatant obtained from lysed B16F10 cells was used to measure nitric oxide (NO), glutathione, and cAMP levels using the NO Assay Kit (Abcam), Glutathione Assay Kit (Abcam), and cAMP ELISA Kit (Enzo Life Sciences, Farmingdale, NY, USA), respectively. The assays were performed according to the manufacturer’s protocols.

### 2.13. Statistical Analysis

Statistical analysis of the experimental results was performed using SPSS (Statistical Package for the Social Sciences, version 28.0, SPSS Inc., Chicago, IL, USA). All data are presented as mean ± standard deviation (SD). All data were analyzed using one-way ANOVA followed by Duncan’s multiple range test to determine significant differences among groups. Different letters indicate statistically significant differences (*p* < 0.05).

## 3. Results

### 3.1. Effects of BRC on the Expression of Skin Moisturizing-Related Genes in UV-Irradiated HaCaT Cells and SKH-1 Hairless Mice

To evaluate the moisturizing and elasticity-enhancing effects of BRC, the expression levels and content of moisturizing-related factors were assessed in UV-irradiated HaCaT cells, a human keratinocyte cell line.

UV irradiation suppressed the expression of HASs (Figure 1A–C), key enzymes involved in hyaluronic acid synthesis, leading to a significant reduction in hyaluronic acid content (Figure 1D). Additionally, UV exposure significantly decreased the mRNA expression of elastin (Figure 1E), LCB1 (SPT) (Figure 1F), and DEGS1 (Figure 1G), which are associated with skin elasticity and ceramide biosynthesis, as well as the intracellular sphingomyelin content (Figure 1H). However, treatment with L-ascorbic acid and BRC significantly restored these levels (*p* < 0.05).

Figure 2 illustrates the effects of UV irradiation and BRC supplementation on the expression of moisturizing-related factors and skin moisture content in SKH-1 hairless mice. Figure 2A presents the morphological and histopathological changes induced by UV exposure, L-ascorbic acid supplementation, and BRC dietary supplementation. L-ascorbic acid and BRC supplementation alleviated various morphological and histopathological alterations caused by UV exposure, including, epidermal thickening, and irregular skin structure. Moreover, skin hydration (Figure 2B) and the expression of related enzymes (Figure 2C–E) were significantly increased in the BRC supplementation groups (*p* < 0.05). Additionally, L-ascorbic acid and BRC supplementation significantly upregulated the mRNA expression of LCB1 (Figure 2F) and DEGS1 (Figure 2G), which are involved in ceramide biosynthesis, as well as fibrillin-1 (Figure 2H), a key protein related to skin elasticity, compared to the control group (*p* < 0.05). These data obtained in this study suggest that BRC supplementation may help mitigate moisture loss and elasticity reduction by effectively inhibiting morphological and histopathological changes as well as oxidative stress induced by UVB irradiation.

### 3.2. Effects of BRC on Skin Barrier Damage and Antioxidant-Related Factors in UV-Irradiated HaCaT Cells and SKH-1 Hairless Mice

UVB irradiation is known to damage the skin barrier and activate NF-κB signaling, leading to inflammatory responses [13]. The induction of inflammation inhibits ceramide synthesis and disrupts the skin barrier [14]. To assess these effects, the expression levels of key proteins in the NF-κB signaling pathway, three pro-inflammatory cytokines, and CerS4 (LASS4) were analyzed. Additionally, to evaluate the antioxidant effects of broccoli extract, the activities of SOD, CAT, and GPx were measured.

UV irradiation stimulated IκB/NF-κB signaling (Figure 3A–C), leading to increased expression of COX-2 (Figure 3D) and key pro-inflammatory cytokines, including TNF-α (Figure 3F), IL-1β (Figure 3G), and IL-6 (Figure 3H), while simultaneously suppressing the expression of CerS4 (Figure 3E). However, treatment with BRC significantly inhibited the activation of NF-κB signaling, resulting in a significant reduction in pro-inflammatory cytokine expression and an increase in CerS4 expression (*p* < 0.05). Meanwhile, to evaluate oxidative stress regulation following UV irradiation, the activities of antioxidant enzymes were measured. SOD (Figure 3I), CAT (Figure 3J), and GPx (Figure 3K) activities were significantly reduced in the UV-irradiated group compared to the normal control group. However, supplementation with L-ascorbic acid and BRC significantly restored these activities, confirming the antioxidant properties of BRC (*p* < 0.05).

Similar results were observed in SKH-1 hairless mice, consistent with the findings from HaCaT cell experiments. UV irradiation activated IκBα/NF-κB signaling (Figure 4A–C), leading to the downregulation of CerS4 (Figure 4E) and increased expression of inflammation-related biomarkers (Figure 4F–H). However, supplementation with L-ascorbic acid and BRC significantly suppressed inflammation and restored CerS4 expression (*p* < 0.05). Furthermore, the activities of SOD, CAT, and GPx (Figure 4I–K), which were reduced due to UV irradiation, were significantly restored following L-ascorbic acid and BRC supplementation (*p* < 0.05). These results indicate that BRC may help protect skin barrier function and maintain hydration by inhibiting UV-induced inflammatory responses and enhancing antioxidant defense mechanisms.

### 3.3. Effects of BRC on Wrinkle Formation-Related Factors in UV-Irradiated HS27 Cell and SKH-1 Hairless Mice

The expression of genes involved in the JNK/c-Fos/c-Jun/MMPs signaling pathway, which is well known to be activated by UV irradiation and plays a key role in wrinkle formation, was analyzed. As shown in Figure 5A, UV irradiation significantly increased JNK, c-Fos, and c-Jun activation (Figure 5B,D) due to elevated ROS production. Additionally, the expression levels of MMP-1, MMP-2, MMP-3, and MMP-9 (Figure 5E–H), which are responsible for extracellular matrix degradation, were markedly increased, leading to a significant reduction in intracellular Collagen I levels (Figure 5I) (*p* < 0.05). Activation of JNK/c-Fos/c-Jun signaling resulted in the suppression of Smad3 phosphorylation (Figure 5J) and TGF-βR1 expression (Figure 5K), along with the downregulation of Procollagen C-endopeptidase enhancer (PCOLCE) and pro-Collagen type I (Figure 5L,M) (*p* < 0.05). However, BRC treatment effectively restored Collagen I levels by modulating the JNK/c-Fos/c-Jun and TGF-βR1/Smad3 signaling pathways (*p* < 0.05).

Similar mechanisms were validated in SKH-1 hairless mice tissues, and the results were consistent with those obtained in HS27 cell experiments. UV irradiation significantly activated the JNK/c-Fos/c-Jun signaling pathway, while suppressing the TGF-βR1/Smad3 signaling pathway (Figure 6J–M), leading to a reduction in Collagen I levels in skin tissues. However, this reduction was reversed by BRC treatment (*p* < 0.05). These findings suggest that BRC may help alleviate UV-induced skin aging and wrinkle formation by inhibiting JNK/c-Fos/c-Jun signaling and enhancing TGF-βR1/Smad3 activation, thereby suppressing collagen degradation and ECM breakdown.

### 3.4. Effects of BRC on Pigmentation-Related Factors in UV-Irradiated B16F10 Cells

To investigate the molecular mechanisms underlying the effects of BRC on melanin production, B16F10 mouse melanoma cells were treated with IBMX to induce melanin synthesis, and the effects of BRC on whitening-related signaling pathways and factors were evaluated (Figure 7). IBMX significantly increased melanin production by activating the cAMP/PKA/CREB signaling pathway. However, treatment with arbutin (positive control) and BRC significantly inhibited the expression of cAMP/PKA/CREB/MITF/TRP-1/TRP-2/tyrosinase pathway proteins and NO, while increasing glutathione levels, ultimately leading to a reduction in melanin content (*p* < 0.05). This study confirms that BRC may effectively reduce melanin accumulation by directly inhibiting melanin production in melanocytes.

## 4. Discussion

UVB and UVA irradiation induce the degradation of ECM components, such as hyaluronic acid and proteoglycans, chronic inflammation, oxidative stress-related cellular damage, DNA damage, and collagen degradation. In particular, UVB exposure generates oxidative stress in cellular components, including lipids, proteins, and DNA, which in turn activates MMPs, contributing to collagen and elastin degradation. As a result, moisturizing factors are depleted, the skin barrier is compromised, leading to reduced skin elasticity, wrinkle formation, and increased melanin synthesis [6,15]. Beyond these systemic effects, NMN is increasingly recognized as a key intermediate in the NAD^+^ biosynthetic pathway that directly contributes to skin homeostasis. Recent evidence suggests that NMN promotes mitochondrial biogenesis and DNA repair in skin cells, enhances antioxidant defenses, and attenuates inflammation through modulation of Sirtuin 1 (NAD-dependent deacetylase sirtuin-1, SIRT1)/NF-κB signaling pathways. These coordinated actions lead to the upregulation of collagen synthesis, reinforcement of skin barrier function, and protection against UV-induced oxidative and inflammatory damage. Nevertheless, despite these promising findings, the precise signaling mechanisms through which NMN mediates skin hydration, elasticity, and barrier improvement remain to be fully elucidated [16,17]. In this study, the effects of BRC (broccoli extract containing more than 99.0% NMN) on UVB-induced skin moisture loss, wrinkle formation, and melanin production, which are key characteristics of photoaging and oxidative stress-induced skin damage, were investigated.

Maintaining an optimal level of skin hydration is essential for preserving epidermal barrier function and overall skin health. This complex process is regulated by the expression and modulation of various genes, each playing a crucial role in the network that governs skin moisture balance. Among these, HAS1-3 are responsible for the synthesis of hyaluronic acid, a key component of skin hydration, while LCB1 and DEGS1 play indispensable roles in sphingolipid metabolism, which is essential for preventing excessive water loss and maintaining skin barrier integrity [18,19,20]. Meanwhile, the deterioration of the skin barrier is significantly influenced by oxidative stress, NF-κB, and COX-2, among other factors. NF-κB, a well-known key transcription factor in inflammatory responses, regulates genes associated with immune and inflammatory reactions, such as pro-inflammatory cytokines, and its activation is commonly observed in skin barrier damage [21,22]. Our study demonstrated that BRC possesses antioxidant properties and enhances skin hydration and barrier function by upregulating hyaluronic acid, HAS1-3, sphingomyelin, elastin, LCB1, DEGS1, fibrillin, and CerS4, while simultaneously suppressing NF-κB signaling and COX-2 expression in HaCaT cells and SKH-1 mice. These findings suggest that BRC contributes to skin hydration and strengthens the epidermal barrier.

The formation of skin wrinkles is intricately linked to the synthesis and degradation of ECM proteins. The TGF-β1/Smad signaling pathway plays a crucial role in collagen synthesis and fibrosis, while the JNK/c-Fos/c-Jun pathway regulates collagen fiber degradation via MMPs. Among these, MMP-1 (collagenase), MMP-2 (gelatinase A), MMP-3 (stromelysin-1), and MMP-9 (gelatinase B) are key enzymes that contribute to the degradation of collagen fibers, leading to the loss of skin elasticity [23,24,25]. The findings of this study demonstrated that BRC prevents wrinkle formation by modulating key pathways involved in collagen metabolism. Specifically, BRC activated the TGF-β1/Smad3 signaling pathway in HS27 cells and SKH-1 mice, leading to increased expression of PCOLCE, a collagen fiber formation enhancer, and pro-collagen type I, a precursor of type I collagen. Additionally, BRC inhibited the JNK/c-Fos/c-Jun/MMPs signaling pathway, thereby suppressing collagen cleavage and degradation, ultimately contributing to wrinkle prevention.

Melanin production is closely associated with UV irradiation, particularly UVB, which directly activates melanocytes and upregulates various melanogenesis-related enzymes, including tyrosinase. The activation of tyrosinase occurs through the cAMP/CREB/MITF/TRPs pathway, a key signaling cascade in melanin synthesis. This pathway facilitates the conversion of tyrosine into melanin, ultimately contributing to melanin production. In addition to the direct activation of melanocytes, UV exposure induces the release of signaling molecules, such as α-melanocyte-stimulating hormone (α-MSH), which further stimulate melanogenesis [26,27,28]. Our study demonstrated that BRC may help reduce pigmentation by inhibiting melanin synthesis through glutathione production and suppressing the cAMP/CREB/MITF/TRPs pathway and tyrosinase activity.

While this study demonstrated that BRC supplementation effectively enhances skin hydration, improves skin barrier function, reduces wrinkle formation, and inhibits melanogenesis, its effects were evaluated in in vitro (HaCaT, HS27, B16F10 cells) and in vivo (SKH-1 hairless mice) models. Although these models provide valuable insights into the molecular mechanisms underlying BRC’s skin-protective effects, they do not fully replicate the complex physiological environment of human skin. One major limitation of this study is the lack of clinical trials confirming the efficacy of BRC in human skin. Therefore, future clinical trials are necessary to validate the beneficial effects of BRC on skin hydration, elasticity, and pigmentation in humans. Despite these limitations, the findings of this study provide a strong foundation for further investigations into the potential use of BRC as a functional ingredient for skin health. Future studies should include long-term evaluations, dose–response studies, and comprehensive clinical trials to confirm the applicability of BRC in functional food formulations.

## 5. Conclusions

This study demonstrated the effects and mechanisms of BRC in alleviating UVB-induced skin dryness, oxidative stress, inflammatory responses, wrinkle formation, and melanin production using both in vitro and in vivo models. BRC promotes skin hydration by enhancing the synthesis of hyaluronic acid and ECM components, including elastin, sphingomyelin, fibrillin, and ceramide. Additionally, it improves skin barrier integrity by reducing oxidative stress and inhibiting the NF-κB signaling pathway and COX-2 expression. Furthermore, BRC effectively prevents wrinkle formation by modulating the JNK/c-Fos/c-Jun/MMPs and TGF-β1/Smad3 signaling pathways and reduces melanin production through the cAMP/PKA/CREB/MITF/TRPs pathway (Figure 8). Based on these findings, we suggest that the consumption of NMN-rich BRC may be beneficial in preventing various types of skin damage caused by photoaging, including moisture loss, barrier impairment, reduced elasticity, and pigmentation.

## Figures and Tables

**Figure 1 cimb-48-00050-f001:**
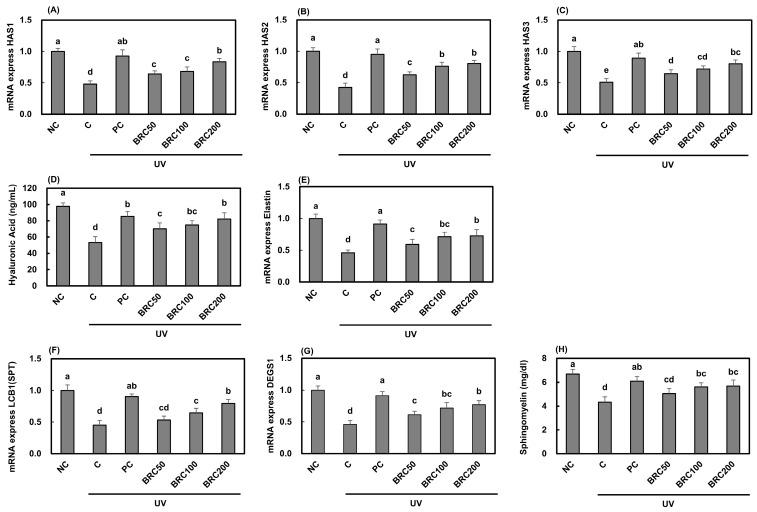
Effects of BRC on the expression of skin hydration-related factors in UV-irradiated HaCaT cells. (**A**–**C**) mRNA expression levels of HAS1, HAS2, and HAS3 were measured using real-time PCR. (**D**) Hyaluronic acid levels were assessed using an ELISA assay. (**E**) mRNA expression of elastin was analyzed by real-time PCR. (**F**,**G**) The expression of sphingolipid metabolism-related genes, LCB1 (SPT) and DEGS1, was determined. (**H**) Sphingomyelin content was evaluated by ELISA. Data are presented as mean ± SD (*n* = 3). Different letters indicate significant differences (*p* < 0.05). NC, normal control; C, UVB irradiation; PC, UVB irradiation + 100 μg/mL of L-ascorbic acid; BRC50, UVB irradiation + 50 μg/mL of BRC; BRC100, UVB irradiation + 100 μg/mL of BRC; BRC200, UVB irradiation + 200 μg/mL of BRC.

**Figure 2 cimb-48-00050-f002:**
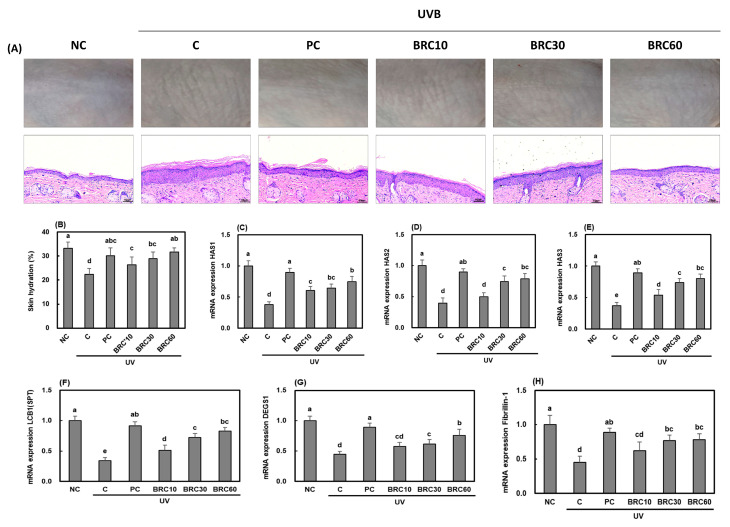
Effects of BRC supplementation on skin hydration-related factors in UV-irradiated SKH-1 hairless mice. (**A**) Morphological changes and histopathological images of skin tissues stained with H&E. (**B**) Skin hydration levels were determined. (**C**–**E**) mRNA expression levels of HAS1, HAS2, and HAS3. (**F**,**G**) Expression of LCB1 and DEGS1 was analyzed. (**H**) mRNA expression levels of fibrillin-1. Data are presented as mean ± SD (*n* = 8). Different letters indicate significant differences (*p* < 0.05). NC, normal control; C, UVB irradiation; PC, UVB irradiation + 100 mg/kg b.w. of L-ascorbic acid; BRC10, UVB irradiation + 10 mg/kg b.w. of BRC; BRC30, UVB irradiation + 30 mg/kg b.w. of BRC; BRC60, UVB irradiation + 60 mg/kg b.w. of BRC.

**Figure 3 cimb-48-00050-f003:**
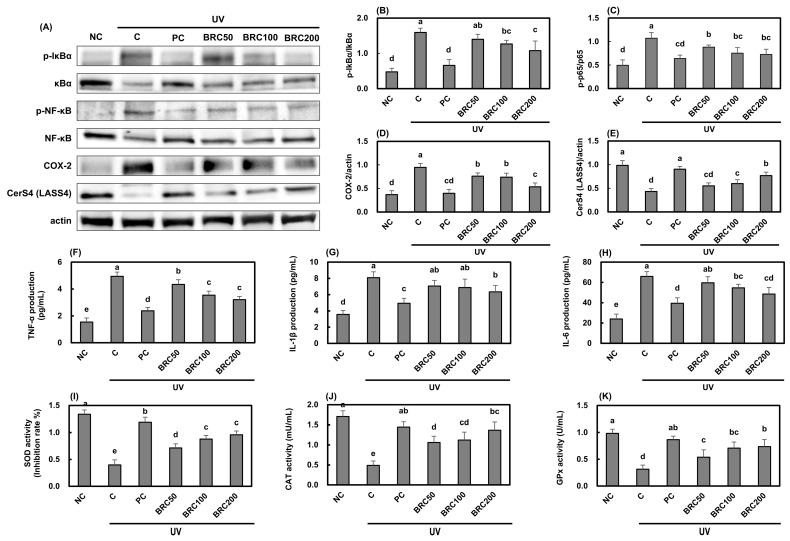
Effects of BRC on skin barrier function and antioxidant enzyme activities in UV-irradiated HaCaT cells. (**A**) Representative Western blot images showing protein levels of NF-κB signaling pathway components (p-IκBα, IκBα, p-NF-κB, NF-κB), COX-2, and CerS4 (LASS4). (**B**–**E**) Quantification of Western blot results for key signaling molecules. (**F**–**H**) Production of pro-inflammatory cytokines TNF-α, IL-1β, and IL-6 measured by ELISA. (**I**–**K**) Activities of antioxidant enzymes SOD, CAT, and GPx. Data are presented as mean ± SD (*n* = 3). Different letters indicate significant differences (*p* < 0.05). NC, normal control; C, UVB irradiation; PC, UVB irradiation + 100 μg/mL of L-ascorbic acid; BRC50, UVB irradiation + 50 μg/mL of BRC; BRC100, UVB irradiation + 100 μg/mL of BRC; BRC200, UVB irradiation + 200 μg/mL of BRC.

**Figure 4 cimb-48-00050-f004:**
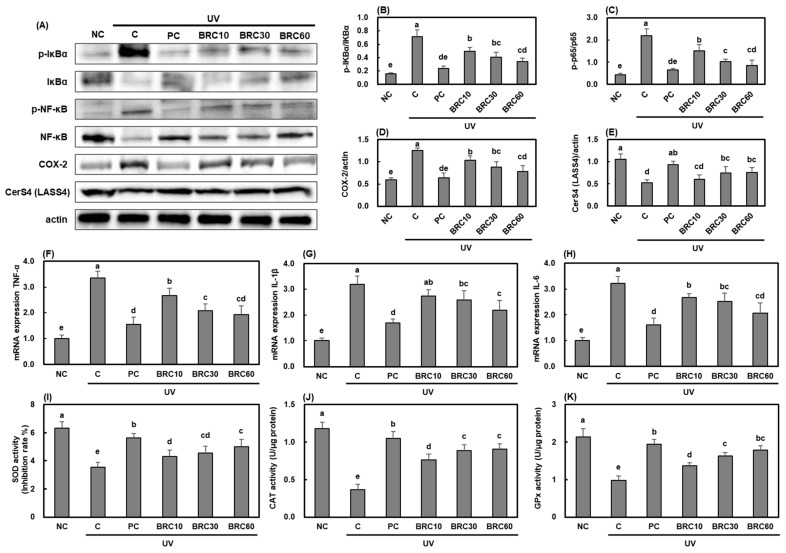
Effects of BRC supplementation on skin barrier function and antioxidant enzyme activities in UV-irradiated SKH-1 hairless mice. (**A**) Western blot images showing NF-κB pathway proteins, COX-2, and CerS4 expression. (**B**–**E**) Quantification of Western blot results. (**F**–**H**) mRNA expression of TNF-α, IL-1β, and IL-6. (**I**–**K**) Antioxidant enzyme activities in skin tissues. Data are presented as mean ± SD (*n* = 8). Different letters indicate significant differences (*p* < 0.05). NC, normal control; C, UVB irradiation; PC, UVB irradiation + 100 mg/kg b.w. of L-ascorbic acid; BRC10, UVB irradiation + 10 mg/kg b.w. of BRC; BRC30, UVB irradiation + 30 mg/kg b.w. of BRC; BRC60, UVB irradiation + 60 mg/kg b.w. of BRC.

**Figure 5 cimb-48-00050-f005:**
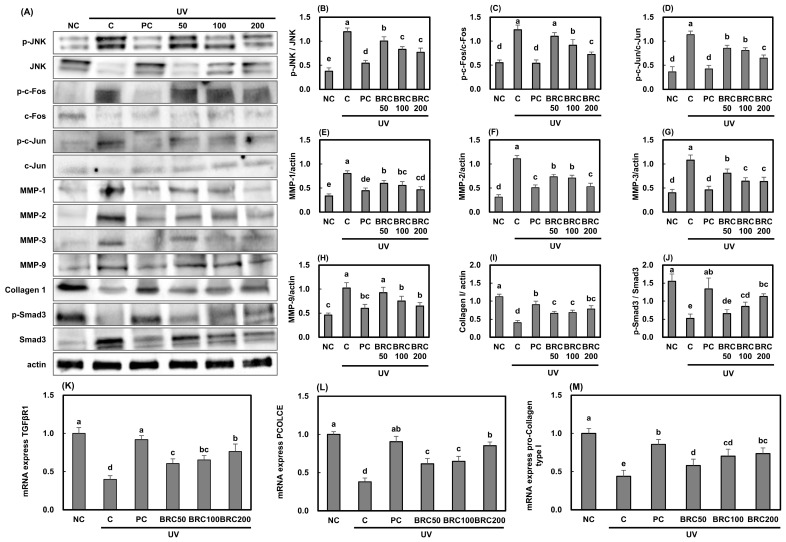
Effects of BRC on wrinkle formation-related signaling pathways in UV-irradiated HS27 cells. (**A**) Western blot images showing JNK/c-Fos/c-Jun and MMP-related protein expression. (**B**–**D**) Quantification of JNK, c-Fos, and c-Jun phosphorylation. (**E**–**H**) Expression of MMP-1, MMP-2, MMP-3, and MMP-9. (**I**) Collagen I levels. (**J**) Phosphorylation of Smad3. (**K**–**M**) mRNA expression of TGF-βR1, PCOLCE, and pro-collagen type I. Data are presented as mean ± SD (*n* = 3). Different letters indicate significant differences (*p* < 0.05). NC, normal control; C, UVB irradiation; PC, UVB irradiation + 100 μg/mL of L-ascorbic acid; BRC50, UVB irradiation + 50 μg/mL of BRC; BRC100, UVB irradiation + 100 μg/mL of BRC; BRC200, UVB irradiation + 200 μg/mL of BRC.

**Figure 6 cimb-48-00050-f006:**
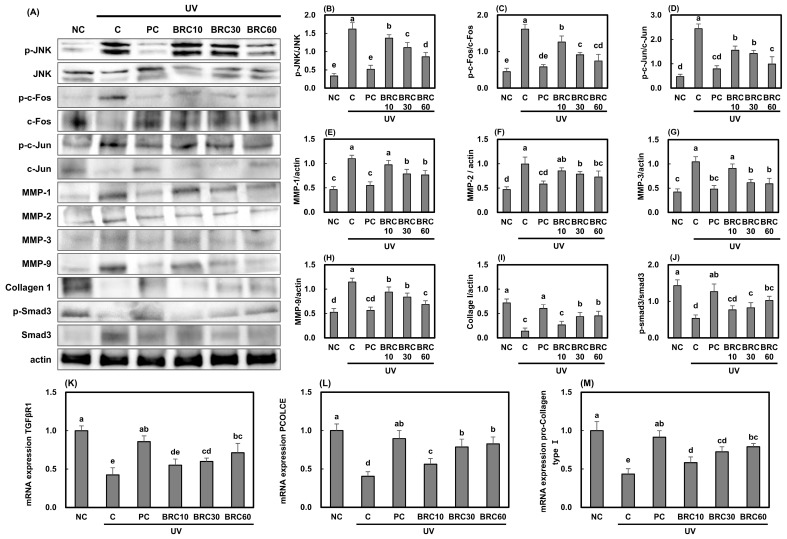
Effects of BRC supplementation on wrinkle formation-related factors in UV-irradiated SKH-1 hairless mice. (**A**) Western blot images showing JNK/c-Fos/c-Jun signaling and MMP expression. (**B**–**D**) Quantification of phosphorylated JNK, c-Fos, and c-Jun. (**E**–**H**) Expression of MMP-1, MMP-2, MMP-3, and MMP-9. (**I**) Collagen I expression. (**J**) Smad3 phosphorylation. (**K**–**M**) mRNA expression of TGF-βR1, PCOLCE, and pro-collagen type I. Data are presented as mean ± SD (*n* = 8). Different letters indicate significant differences (*p* < 0.05). NC, normal control; C, UVB irradiation; PC, UVB irradiation + 100 mg/kg b.w. of L-ascorbic acid; BRC10, UVB irradiation + 10 mg/kg b.w. of BRC; BRC30, UVB irradiation + 30 mg/kg b.w. of BRC; BRC60, UVB irradiation + 60 mg/kg b.w. of BRC.

**Figure 7 cimb-48-00050-f007:**
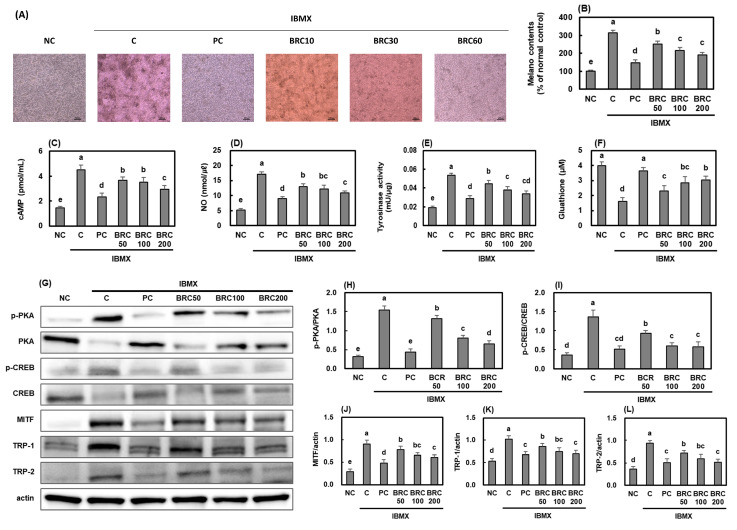
Effects of BRC on melanogenesis-related signaling pathways in IBMX-stimulated B16F10 cells. (**A**) Representative images showing melanin accumulation. (**B**) Quantification of melanin content. (**C**–**F**) Levels of cAMP, NO, tyrosinase activity, and glutathione. (**G**) Western blot images showing cAMP/PKA/CREB/MITF/TRPs pathway proteins. (**H**–**L**) Quantification of phosphorylated PKA, CREB, MITF, TRP-1, and TRP-2 expression levels. Data are presented as mean ± SD (*n* = 3). Different letters indicate significant differences (*p* < 0.05). NC, normal control; C, UVB irradiation; PC, UVB irradiation + 100 μg/mL of arbutin; BRC50, UVB irradiation + 50 μg/mL of BRC; BRC100, UVB irradiation + 100 μg/mL of BRC; BRC200, UVB irradiation + 200 μg/mL of BRC.

**Figure 8 cimb-48-00050-f008:**
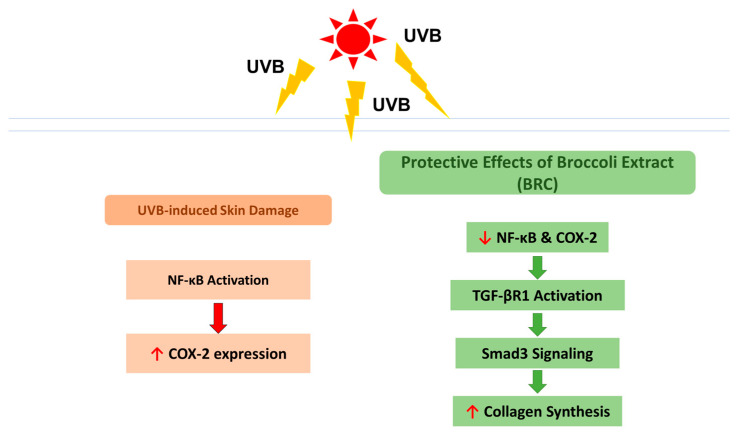
Schematic illustration of the protective mechanisms of broccoli extract (BRC) against UVB-induced skin damage.

**Table 1 cimb-48-00050-t001:** Primer sequences used in real-time PCR quantification of mRNA.

Gene	Primer Sequences
HAS1 ^1^ (H)	F 5′-ATG TGG AGC GGG CTT GTC-3′ R 5′-AGG CCT AGA GGA CCG CTG AT-3′
HAS2 (H)	F 5′-GAA ACA GCC CCA GCC AAA-3′ R 5′-AAG ACT CAG CAG AAC CCA GGA A-3′
HAS3 (H)	F 5′-TGC TTG CCC TCC AAA TGT C-3′ R 5′-CCT CTT GTC TGC TGT CCA CCT T-3′
DEGS1 ^2^ (H)	F 5′-GCT GAT GGC GTC GAT GTA GA-3′ R 5′-TGA AAG CGG TAC AGA AGA ACC A-3′
Elastin (H)	F 5′-GTC GGA GTC GGA GGT ATC-3′ R 5′-TGA GAA GAG CAA ACT GGG-3′
TGF-βR1 ^3^ (H)	F 5′-TCC CGG CAG ATC AAC GA-3′ R 5′-ACG CGG TCA CAA ACA TGG T-3′
PCOLCE ^4^ (H)	F 5′-TTA CGT GGC AAG TGA GGG TTT-3′ R 5′-TGT CCA GAT GCA CTT CTT GTT TG-3′
Pro-Collagen type I (H)	F 5′-GAC CGT TCT ATT CCT CAG TGC AA-3′ R 5′-CCC GGT GAC ACA CAA AGA CA-3′
GAPDH (H)	F 5′-CCC CAC ACA CAT GCA CTT ACC-3′ R 5′-TTG CCA AGT TGC CTG TCC TT-3′

^1^ Hyaluronic acid synthase, ^2^ Delta 4-desaturase, sphingolipid 1, ^3^ Transforming growth factor-beta receptor 1, ^4^ Procollagen C-endopeptidase enhancer.

**Table 2 cimb-48-00050-t002:** Primer sequences used in RT-PCR quantification of mRNA.

Gene	Primer Sequences
HAS1 ^1^ (M)	F 5′-TCA GGG AGT GGG ATT GTA GGA-3′ R 5′-AAA TAG CAA CAG GGA GAA AAT GGA-3′
HAS2 (M)	F 5′-AAT ACA CGG CTC GGT CCA AGT-3′ R 5′-CCA TCG GGT CTG CTG GTT-3′
HAS3 (M)	F 5′-GGC CAT GGG AGC TAA AGT TG-3′ R 5′-CCA AAT TGA TGT TGA AAC TCT TGA AA-3′
LCB1 (SPT) ^2^ (M)	F 5′-AGC GCC TGG CAA AGT TTA TG-3′ R 5′-GTG GAG AAG CCG TAC GTG TAA AT-3′
DEGS1 ^3^ (M)	F 5′-CCG GCG CAA GGA GAT CT-3′ R 5′-TGT GGT CAG GTT TCA TCA AGG A-3′
Fibrillin-1 (M)	F 5′-ACA ATT GTT CAC CGA GTC GAT CT-3′ R 5′-ACT GTA CCT GGG TGT TGC CAT T-3′
TNF-α ^4^ (M)	F 5′-ACC CCC CCA TGC TAA GTT CT-3′ R 5′-ATG CCT GTG TCT ATT TCC TTT TGA T-3′
IL-1β ^5^ (M)	F 5′-GTC GCT CAG GGT CAC AAG AAA-3′ R 5′-AAG GAG GAA AAC ACA GGC TCT CT-3′
IL-6 (M)	F 5′-CCA CGG CCT TCC CAT CTT C-3′ R 5′-TTG GGA GTG GTA TCC TCT GTG A-3′
TGF-βR1 ^6^ (M)	F 5′-CATCCTGATGGCAAGAGCTACA-3′ R 5′-TAGTGGATGCGGACGTAACCA-3′
PCOLCE ^7^ (M)	F 5′-TTA CGT GGC AAG TGA GGG TTT-3′ R 5′-TGT CCA GAT GCA CTT CTT GTT TG-3′
Pro-collagen type 1 (M)	F 5′-GAC CGT TCT ATT CCT CAG TGC AA-3′ R 5′-CCC GGT GAC ACA CAA AGA CA-3′
GAPDH (M)	F 5′-CAT GGC CTT CCG TGT TCC TA-3′ R 5′-GCG GCA CGT CAG ATC CA-3′

^1^ Hyaluronic acid synthase, ^2^ Long chain base biosynthesis protein 1, ^3^ Delta 4-desaturase, sphingolipid 1, ^4^ Tumor necrosis factor, ^5^ Interleukin, ^6^ Transforming growth factor-beta receptor 1, ^7^ Procollagen C-endopeptidase enhancer.

## Data Availability

The original contributions presented in this study are included in the article. Further inquiries can be directed to the corresponding authors.

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
