# Peer review of "Curr. Issues Mol. Biol.2026, 48(1), 50;https://doi.org/10.3390/cimb48010050"

_cimb, 2025, doi:10.3390/cimb48010050_

Round 1

Reviewer 1 Report

Comments and Suggestions for Authors

Dear Authors,

Your original scientific manuscript “Broccoli Extract (Broccoli NMN®) Improves Skin Hydration by 2 Regulating HAS and NF-κB Pathways and Reduces Wrinkle 3 Formation via the TGF-βR1/Smad3/Collagen Pathway” deals with less research mechanistical insight behind the already well-known health beneficial effects of broccoli extract, which has a huge potential.

Your work emphasis a step beyond the observed effects, by providing molecular understanding of signaling processes behind moisture loss, inflammation, wrinkle formation and melanin production. There are not many papers on that topic, and therefore new insights you are presenting would be valuable for the Scientific Community in the field and the readers of the Current Issues in Molecular Biology. Especially, since your work offers in vitro and in vivo study, which supports potential of such extract for further clinical trials. Therefore, my recommendation is accepting after minor revision, after revising a manuscript according to reviewers’ comments and suggestions.

General comment would be to add into Introduction a few sentences about already known beneficial effects of broccoli extract specifically related to the skin health, and some references on that topic.

Questions would be about the Broccoli extract (BRC) which was obtained from Daedeok Pharma Co., Ltd. Do you know how it is obtained, by which solvent? Is it an extract or a purified compound since it is said in 2.1.“ it is characterized and containing 99.0% or more β-NMN”? Also, in the conclusion you say: “One major limitation of this study is the lack of clinical trials confirming the efficacy of BRC in human skin” (Ln 435-436), so I wonder is it for "Research only", or is it already on the market? For what is this extract used, since it is obtained from Daedeok Pharma Co., Ltd. Do they have some clinical data about it?

Specific comments:

Pg. 2, Ln 82: I would suggest a change of subtitle 2.1. Preparation of Broccoli extract (BRC), since you have purchase it. Also, regarding my upper question I suggest providing some additional information about extract itself?

Pg.2., Ln 59-64 – a repetition of the sentence!

Pg. 3, Ln 100-101 you have already introduced abbreviation for broccoli extract (BRC) so you do not need that here

Author Response

We appreciate the reviewer’s insightful and constructive comments regarding the broccoli extract (BRC) used in this study. Our detailed responses are provided below.

Questions were raised regarding the origin, extraction method, solvent used, and purity of the broccoli extract (BRC) supplied by Daedeok Pharma Co., Ltd. In particular, clarification was requested as to whether BRC is an extract or a purified compound, given that Section 2.1 states it contains ≥99.0% β-NMN. Additionally, the reviewer inquired whether BRC is for research use only or commercially available, its intended application, and whether any clinical data exist, especially in light of the limitation stated in the Conclusion regarding the lack of clinical trials.

  • The broccoli extract (BRC) used in this study was investigated as a functional food ingredient with potential benefits for skin health. This research was designed as a basic preclinical investigation, and the efficacy of BRC was evaluated through in vitro and in vivo non-clinical experiments. Based on the promising outcomes observed in this study, clinical trials are planned for future research to further validate its effects on human skin. At the time of this study, BRC was supplied by Daedeok Pharma Co., Ltd. specifically for research purposes and has not yet been evaluated through human clinical trials related to skin health. Therefore, as stated in the Conclusion section, the absence of clinical evidence represents a limitation of the present study. To the best of our knowledge, no clinical data regarding skin-related outcomes of BRC are currently available. Future clinical investigations are planned to address this limitation. In response to the reviewer’s request for clarification, we have revised the Methods section to include more detailed information regarding the extraction and purification processes of BRC. Although the material was purchased, the description has been expanded to clarify that BRC is a broccoli-derived extract that has undergone additional purification steps to achieve a high content (≥99.0%) of β-NMN.

Pg. 2, Ln 82: I would suggest a change of subtitle 2.1. Preparation of Broccoli extract (BRC), since you have purchase it. Also, regarding my upper question I suggest providing some additional information about extract itself?

  • In response to the reviewer’s comment, we have revised the Methods section to include detailed information on the extraction and purification procedures of the broccoli extract (BRC). Accordingly, the description of the extract has been expanded to clarify its preparation process, even though the material was purchased.

Pg.2., Ln 59-64 – a repetition of the sentence!

  • The repeated sentence has been deleted accordingly. We thank the reviewer for this careful observation.

Pg. 3, Ln 100-101 you have already introduced abbreviation for broccoli extract (BRC) so you do not need that here

  • The redundant definition of the abbreviation for broccoli extract (BRC) in Lines 100–101 has been removed in the revised manuscript. We thank the reviewer for this helpful comment.

Reviewer 2 Report

Comments and Suggestions for Authors

The manuscript investigates the protective efficacy and mechanism of Broccoli extract (BRC) against UVB-induced skin damage. Results indicated that BRC alleviated UVB-induced skin dehydration by suppressing NF-κB signaling and COX-2 expression. Furthermore, BRC promoted collagen synthesis by activating TGF-βR1/Smad3/Collagen pathway. However, several major concerns need to be addressed to strengthen the validity and impact of the study.

(1) Figure 2A and figure 7A: The scale bars in histology images should be presented in all figures.

(2) The mechanisms by which BRC prevent UVB-induced skin damage is complicated. It is suggested to provide a schematic summary figure.

(3) This manuscript was focused on the exploration of β-NMN (purity > 99.0%) against skin damage. β-NMN is widely presented in the nature. The title of this manuscript should be revised.

(4) Typing and labelling errors should be corrected.

Author Response

We thank the reviewer for the thorough evaluation of our manuscript and for the valuable comments, which have helped us to improve the clarity and impact of the study. Our point-by-point responses are provided below.

1) Figure 2A and figure 7A: The scale bars in histology images should be presented in all figures.

-> Scale bars were already provided in Figure 2A. In response to the reviewer’s comment, scale bars have now been added to Figure 7A in the revised manuscript.

2) The mechanisms by which BRC prevent UVB-induced skin damage is complicated. It is suggested to provide a schematic summary figure.

-> We appreciate this helpful suggestion. To improve clarity and provide an integrated overview of the proposed mechanisms, a schematic summary figure has been added as Figure 8 in the Conclusion section of the revised manuscript. This figure summarizes the protective effects of BRC against UVB-induced skin damage, including the suppression of NF-κB/COX-2–mediated inflammatory signaling and the activation of the TGF-βR1/Smad3/collagen synthesis pathway, thereby facilitating a clearer understanding of the overall mechanism of action.

3) This manuscript was focused on the exploration of β-NMN (purity > 99.0%) against skin damage. β-NMN is widely presented in the nature. The title of this manuscript should be revised.

->  As described in the Methods section, the β-NMN used in this study was obtained from broccoli extract through multiple purification steps to enhance its purity, rather than being a synthetically isolated or naturally occurring free β-NMN. Therefore, this study focuses on the biological effects of a broccoli-derived extract enriched in β-NMN, and we believe that revising the title to reflect β-NMN alone would not accurately represent the nature of the experimental material used in this work.

4. Typing and labelling errors should be corrected.

-> In addition, all identified typing and labeling errors have been carefully corrected in the revised manuscript.

Reviewer 3 Report

Comments and Suggestions for Authors

The manuscript presents a comprehensive description of the purported effects of BRC on UVB-induced skin damage; however, several claims appear overly conclusive and would benefit from a more cautious interpretation. Although the study outlines multiple mechanisms—ranging from oxidative-stress reduction to modulation of various signaling pathways—the evidence supporting these mechanistic assertions is not clearly detailed in the current text. It would be advisable for the authors to provide more explicit data demonstrating how each pathway was validated, as well as the extent to which the findings from in vitro and in vivo models can be generalized. Furthermore, statements suggesting that NMN-rich BRC consumption may prevent diverse forms of photoaging should be tempered unless supported by robust clinical evidence. Overall, the section would benefit from clearer methodological transparency and a more balanced presentation of the results.

Please include a metric scale bar in the histological photographs shown in Figures 2 and 7 to provide an appropriate reference for size and magnification.

Author Response

We sincerely thank the reviewer for the thoughtful and constructive comments, which have greatly contributed to improving the rigor and clarity of this manuscript.

In response, we have carefully revised the manuscript to adopt a more cautious and balanced interpretation of the findings. Statements that could be interpreted as overly conclusive—particularly those suggesting that NMN-rich BRC consumption may broadly prevent diverse forms of photoaging—have been moderated to ensure that the conclusions are appropriately aligned with the preclinical nature of this study.

To enhance methodological transparency, we have clarified the experimental evidence supporting each proposed mechanism, including oxidative stress reduction and the modulation of specific signaling pathways. These mechanistic claims are now more explicitly linked to the corresponding in vitro and in vivo results, with clearer references to the relevant figures and analyses. In addition, we have emphasized the limitations regarding the generalizability of the findings from cellular and animal models and have clearly stated that further well-designed clinical studies are required to validate the effects of NMN-rich BRC on human skin.

Regarding the histological images, metric scale bars were already included in Figure 2A. In response to the reviewer’s comment, scale bars have now been added to Figure 7A in the revised manuscript to provide appropriate references for size and magnification.

We believe that these revisions have improved the overall balance, transparency, and scientific rigor of the manuscript.